# Interpretation of PSMA-PET Among Urologists: A Prospective Multicentric Evaluation

**DOI:** 10.3390/cancers17132122

**Published:** 2025-06-24

**Authors:** Guglielmo Mantica, Francesco Chierigo, Francesca Ambrosini, Francesca D’Amico, Greta Celesti, Arianna Ferrari, Fabrizio Gallo, Maurizio Schenone, Andrea Benelli, Carlo Introini, Rosario Leonardi, Alessandro Calarco, Francesco Esperto, Andrea Pacchetti, Rocco Papalia, Giorgio Bozzini, Armando Serao, Valentina Pau, Gianmario Sambuceti, Carlo Terrone, Giuseppe Fornarini, Matteo Bauckneht

**Affiliations:** 1Department of Surgical and Diagnostic Integrated Sciences (DISC), University of Genova, 16126 Genova, Italy; ariannaf1996@gmail.com (A.F.); valentina.pau@hsanmartino.it (V.P.); calro.terrone@hsanmartino.it (C.T.); 2IRCCS Ospedale Policlinico San Martino, 16132 Genova, Italy; francesca.ambrosini@hsanmartino.it (F.A.); gianmario.sambuceti@unige.it (G.S.); giuseppe.fornarini@hsanmartino.it (G.F.); matteo.bauckneht@hsanmartino.it (M.B.); 3Department of Urology, ASST Santi Paolo e Carlo, 20153 Milan, Italy; francesco.chierigo@gmail.com; 4Department of Health Sciences (DISSAL), University of Genoa, 16126 Genova, Italy; francesca.damico@hsanmartino.it (F.D.); greta.celestri@hsanmartino.it (G.C.); 5Nuclear Medicine, Department of Biomedical and Dental Sciences and Morpho-Functional Imaging, University of Messina, 98125 Messina, Italy; 6Department of Urology, San Paolo Hospital, 17100 Savona, Italy; fabriziogallo.sv@gmail.com (F.G.); m.schenone@asl2.liguria.it (M.S.); 7E.O. Ospedali Galliera, UOC Urologia, 16128 Genova, Italy; andrea.benelli@galliera.it (A.B.); carlo.introini@galliera.it (C.I.); 8Casa di Cura Musumeci-Gecas, 95030 Gravina di Catania, Italy; rosario.leonardi@unikore.it; 9Department of Urology, San Carlo di Nancy Hospital, 00165 Rome, Italy; alecalarco@gmail.com; 10Department of Urology, Fondazione Campus Bio-Medico of Rome, 00128 Rome, Italy; francescoesperto@gmail.com (F.E.); rocco.papalia@unicampus.it (R.P.); 11Department of Urology, ASST Lariana, Sant’Anna Hospital, San Fermo della Battaglia, 22042 Como, Italy; a.pacchetti.90@gmail.com (A.P.); gioboz@yahoo.it (G.B.); 12Urologia, AO SS Antonio e Biagio e Cesare Arrigo, 15121 Alessandria, Italy; armando.serao@gmail.com

**Keywords:** PSMA-PET, prostate cancer, interobserver agreement, PCa

## Abstract

This study evaluated how accurately urologists can interpret PSMA-PET scans for staging high-risk prostate cancer. Sixty-three urologists from eight Italian centers assessed 20 scans without clinical context. Results showed that interpretation is difficult: only one participant achieved full accuracy, and less than half reached an acceptable level for partial staging. Urologists from high-volume centers and those involved in multidisciplinary teams performed better, suggesting that experience and collaboration improve diagnostic skills. These findings underscore the importance of structured training and multidisciplinary exposure to enhance staging accuracy and ultimately improve patient outcomes.

## 1. Introduction

Prostate cancer (PCa) remains one of the most prevalent malignancies among men, with a wide spectrum of clinical behavior ranging from indolent to highly aggressive forms [1]. The accurate staging and assessment of the disease extent are crucial in tailoring treatment strategies and improving patient outcomes [2,3,4]. In recent years, the advent of prostate-specific membrane antigen (PSMA) positron emission tomography (PET) has revolutionized the diagnostic and staging approach in prostate cancer [5], providing enhanced sensitivity for the detection of localized and metastatic disease. PSMA-PET, due to its superior imaging capabilities, has significantly improved the ability to identify subclinical lesions and metastases, even in cases of biochemical recurrence or advanced disease [6]. According to the current international guidelines, including those from the European Association of Urology (EAU) and the American Urological Association (AUA), PSMA-PET/CT is recommended for staging in patients with unfavorable intermediate-, high-, and very high-risk prostate cancer and for the detection of biochemical recurrence after radical treatment when conventional imaging is inconclusive or negative [7,8].

However, the successful integration of PSMA-PET into clinical practice requires urologists to be proficient not only in its indications and utility but also in interpreting its results correctly. The ability to analyze PSMA-PET scans is essential for informed decision-making, particularly in the context of precision medicine, where tailored therapeutic interventions are paramount. This article focuses on the interpretation capability regarding this advanced imaging technique among a wide range of urologist specialists (either fully formed or in training), with the final aim to optimize patient care.

The interpretation of PSMA-PET is challenging even for experienced nuclear medicine specialists. However, to the best of our knowledge, no previous study has addressed the ability of urologists to identify suspicious lesions in PSMA-PET. To address this gap, we relied on a multicentric prospective survey at eight Italian institutions.

## 2. Materials and Methods

Challenging images of 20 patients submitted to PSMA-PET for the primary staging of high-risk prostate cancer at IRCCS Ospedale Policlinico San Martino of Genova between February 2021 and February 2024 were selected by an expert uro-nuclear medicine specialist (M.B.). In ten cases, PSMA-PET was performed using [68Ga] Ga-PSMA-11, while, in the remaining cases, the used tracer was [18F] PSMA-1007. Images were acquired following the current guidelines. All studies were performed using state-of-the-art equipment, including the Biograph Hirez 16 (Siemens Healthineers, Erlangen, Germany), Biograph MCT Flow (Siemens Healthineers), and Omni Legend 32 (GE Healthcare). Appendix A provides the correct staging of the cases.

Two nuclear medicine physicians not involved in the selection process (F.D.A. and G.C.) reviewed the PSMA-PET scans and defined a composite reference standard by consensus according to the previously described methodology [8,9]. PSMA-PET images were shown to urologists (residents or consultants with at least three years of clinical experience) from eight institutions (six academic and two non-academic) from June 2024 to December 2024. The study adhered to the guidelines of the Declaration of Helsinki and was approved by the local ethical committee (registration number 5/2023—DB ID 12914). All subjects gave written informed consent.

### 2.1. Test Administration

PSMA-PET was administered using the RadiAnt DICOM Viewer 4.0.1 software (Medixant, Poznan, Poland). Every participant had a maximum of 10 min to evaluate each PSMA-PET. First, each participant answered an anonymous survey (see Appendix A) with demographics and professional data. Second, participants were asked to identify any prostatic, lymph-nodal, and metastatic involvement, whenever present. No clinical data were given to the readers so as not to influence the outcomes of the PSMA-PET interpretation. Eight “presenters” (A.F., F.E., A.C., A.B., F.C., A.P., R.L., F.G.) administered the test in the seven different centers.

### 2.2. Statistical Analysis

Continuous variables were summarized as medians and interquartile ranges (IQRs) and categorical variables as the number of subjects and percentage values. Statistical analyses were based on two steps. First, data were analyzed according to urologists’ hierarchy (resident vs. consultant), the type of institution (academic vs. non-academic), the number of prostate cancer (PCa) diagnoses per year per institution (≥300 vs. <300), and the number of robot-assisted radical prostatectomies (RARPs) per year per institution (≥150 vs. <150), as well as according to the presence of a PCa multidisciplinary team (MDT). The Wilcoxon rank sum test, Pearson’s chi-squared test, and Fisher’s exact test were used where appropriate for this analysis. Second, we aimed to assess potential predictors of proficiency in PSMA-PET reading, defined as achieving a 75% cut-off in the correct identification of T, N, and M stages in the 20 cases, at least 2 out of 3 stages among the T, N, and M stages of every single case (almost correct cases), and the complete stage of the case (correct case). For this purpose, we relied on univariable and multivariable logistic regression models. The covariates of the multivariable logistic regression models consisted of the hierarchy (resident vs. consultant), number of robot-assisted radical prostatectomies (RARPs) per year per institution (≥150 vs. <150), and presence of a PCa MDT. For all statistical analyses, the R software environment for statistical computing and graphics (version 3.4.3, R Foundation, Vienna, Austria) was used. All tests were 2-sided, with the level of significance set at *p* = 0.05.

## 3. Results

Overall, we collected complete data for 63 urologists, with a median (IQR) age of 36 (30, 42) years. Of these, 26 (41%) and 37 (59%) were, respectively, residents and consultants (Table 1). Most of the responders (52 participants, 83%) were directly involved in the diagnosis and management of PCa. In total, 68% of the participants had experience in prostate biopsy, while most of them (49%) had experience with MRI fusion biopsy. In three centers (18 participants, 29%), less than 300 PCa diagnoses a year were performed; in five centers (35 participants, 56%), less than 150 RARPs a year were performed; and in four centers (23 participants, 37%), PCa MDTs were not part of routine patient care. The median (IQR) number of correct cases was five (four, seven), with only one responder reaching the 75% cut-off of correct responses. When lowering the threshold to correctly identify at least two out of three components of the staging [local staging (T stage), regional lymph node metastases (N stage), and distant metastases (M stage)], the median (IQR) number of correct cases was 14 (13, 16), with 44% of responders achieving the 75% cut-off.

Looking at the single regions of the staging, the median (IQR) number of correct T stages was 11 (10, 15), with 19 (30%) responders achieving proficiency; the median (IQR) number of correct N stages was 14 (13, 16), with 30 (48%) responders achieving proficiency; and the median (IQR) number of correct M stages was 12 (11, 14), with 10 (16%) responders achieving proficiency. In our cohort of responders, no statistically significant differences were recorded according to the type of institution (academic vs. non-academic) or hierarchy (consultant vs. resident, Table 2). However, an analysis according to the number of PCa diagnoses revealed that responders in centers with ≥300 diagnoses per year had more correct T and M stage identifications [11 (10, 16) vs. 11 (9, 13), *p* = 0.032, and 13 (12, 14) vs. 11 (10, 16), *p* = 0.002, respectively], with more T and M stage proficiency [17 (38%) vs. 2 (11%), *p* = 0.037, and 10 (22%) vs. 0 (0%), *p* = 0.050, respectively] and more correct cases [6 (4, 9) vs. 4 (2,6), *p* = 0.008]. Similarly, responders from centers performing ≥150 RARPs per year correctly identified the M stage more often than their counterparts [13 (12, 14) vs. 12 (11, 13), *p* = 0.030], as well as correctly staging more cases [6 (4, 10) vs. 5 (3, 6), *p* = 0.042]. Finally, responders from institutions with a PCa MDT correctly diagnosed more distant metastases [13 (12, 14) vs. 12 (11, 13), *p* = 0.026]. In the univariable logistic regression analyses, the hierarchy, type of institution, number of PCa diagnoses per year per institution, number of RARPs per year per institution, personal experience in standard or fusion prostate biopsy, involvement in the diagnosis and management of PCa, and presence of a PCa MDT were not predictors of proficiency in T, N, and M stage identification or in almost correct staging. Analyses for correct case staging were not performed as only one responder achieved the 75% cut-off, as previously mentioned. In the multivariable logistic regression analyses (Table 3), no statistically significant predictor of proficiency in T, N, and M stage identification was found, nor in almost correct staging (Table 3A–D). However, in the proficiency of identifying lymph node metastases, the hierarchy and number of RARPs per year per institution almost reached the level of significance. Specifically, residents were found to be less likely to proficiently identify nodal metastases (OR 0.31, 95%CI 0.09, 1.02, *p* = 0.062) and responders from institutions performing more than 150 RARPs per year were found to be more likely to proficiently identify nodal metastases (OR 2.98, 95% CI 0.99, 9.61, *p* = 0.056), although this was not statistically significant.

## 4. Discussion

In this study, the ability of urologists to interpret PSMA-PET images was investigated. To our knowledge, this is the first study to examine urologists’ skills in interpreting PSMA-PET.

Our analysis found that urologists working in high-volume centers, particularly those diagnosing more than 300 prostate cancer cases per year, demonstrated greater proficiency in determining the T and M stages. Similarly, those from institutions performing ≥150 RARPs per year were more likely to correctly identify the M stage than their counterparts. Increased exposure to a broad spectrum of cases may improve clinicians’ ability to effectively interpret PSMA-PET scans.

Interestingly, our multivariable logistic regression analyses revealed no statistically significant predictors of the ability to perform staging (T, N, or M) or near-correct staging. This suggests that PSMA-PET interpretation remains a difficult task for urologists, regardless of their level of experience or institutional background, emphasizing the need for structured training in PSMA-PET imaging interpretation.

Despite the lack of statistically significant predictors, it is noteworthy that the hierarchy (resident vs. consultant) and institutional surgical volume (≥150 RARPs per year) reached near significance in predicting proficiency in identifying lymph node metastases. This trend suggests that greater surgical exposure and experience in clinical decision-making may contribute to the better recognition of nodal disease patterns on PSMA-PET.

Residents were less likely to correctly recognize lymph node metastases (OR 0.31, 95%CI 0.09–1.02, *p* = 0.062), possibly due to their inevitably limited experience in imaging interpretation and multidisciplinary discussions compared to specialists. This highlights the importance of the early involvement of trainees in multidisciplinary teams (MDTs) and imaging discussions to enhance their diagnostic skills [10,11].

Similarly, responders from high-volume surgical centers (≥150 RARPs per year) were more likely to proficiently identify nodal metastases (OR 2.98, 95% CI 0.99–9.61, *p* = 0.056). Although this result was not statistically significant, it suggests that urologists who are frequently confronted with prostate cancer cases are better able to detect nodules on PSMA-PET. Although statistical significance was not reached, these results emphasize the potential benefit of surgical interventions and high case numbers in developing skills in PSMA-PET interpretation. Future studies with larger case numbers could further clarify the relationship between institutional factors, experience levels, and staging accuracy.

The observed gaps in PSMA-PET interpretation skills among urologists—especially residents and those outside academic centers—underline the need for targeted educational interventions, such as dedicated imaging modules or case-based training within urology residency programs.

Our results show that the most critical challenges in PSMA-PET interpretation are related to local tumor staging (T stage) and distant metastases (M stage), which are crucial for treatment decisions.

One of the main problems at the T stage appears to be related to the biodistribution properties of [68Ga]Ga-PSMA-11, which is primarily excreted through the urinary tract [10]. This physiological excretion can lead to high background activity in the bladder and urethra, making it difficult to distinguish primary tumors or local extensions. This is a well-documented limitation, and its impact on the staging accuracy is particularly relevant when specialists with limited experience in nuclear imaging interpret PSMA-PET.

Moreover, several studies have demonstrated the increased uptake of PSMA tracers in the ganglia of the sympathetic trunk, which may introduce another cause of misinterpretation [12,13,14,15,16]. These uptakes can represent a diagnostic pitfall in the interpretation of PSMA-PET imaging, as they may be misinterpreted as metastases to non-regional lymph nodes.

Similarly, our data suggest that M staging is another major challenge, likely due to the intrinsic properties of [18F]PSMA-1007. In contrast to [68Ga]Ga-PSMA-11, this tracer is excreted in the urine to a lesser extent but has a higher incidence of unspecific bone uptake (UBU). UBU refers to regions with increased tracer activity within the bone, without a clear benign or malignant origin, most frequently observed in the ribs and pelvis [17]. Notably, UBU incidence is not a peculiarity of [18F]PSMA-1007; however, this tracer is the one that shows the highest risk for UBU occurrence [17]. Our analysis showed that participants who worked in facilities with a dedicated multidisciplinary team (MDT) for prostate cancer had better accuracy in detecting distant metastases. This finding underscores the importance of shared decision-making and cumulative experience in enhancing diagnostic skills. Regular exposure to PSMA-PET scans through MDT discussions likely enhances clinicians’ ability to distinguish between true-positive and false-positive findings. This finding is consistent with previous studies that have shown a correlation between greater PSMA-PET reporting experience and improved diagnostic accuracy, as well as enhanced patient management [18,19,20].

To date, there is limited literature addressing the ability of clinicians who are not radiologists to interpret imaging studies independently, although, in routine clinical practice, physicians often not only read the official radiological report but also review the images themselves. While some studies have investigated clinicians’ preferences for structured report formats, little is known about their actual interpretation skills, which are an important but under-researched component of decision-making in multidisciplinary care [21].

Despite the novelty of this study, several limitations must be acknowledged. First, the interpretation of PSMA-PET scans was performed without the availability of clinical information, which may not fully reflect real-life conditions, where medical history and additional imaging findings aid in diagnostic interpretation. Second, the study was limited to a single country and included a relatively small number of institutions, which may affect the generalizability of the results. Third, the definition of competence was based on an arbitrarily chosen cut-off of 75%, and further research is needed to establish standardized measures of competence for non-nuclear medicine physicians in reading PSMA-PET. We selected this value as a commonly used benchmark in the educational literature to indicate a minimum level of acceptable competence. Future research might explore alternative or more nuanced measures of proficiency (e.g., item-level difficulty analysis, expert consensus benchmarks, etc.).

Moreover, the absence of statistically significant associations in several regression models may have limited the statistical power, particularly given the modest sample size. The exploratory nature of the study and the available sample size may have limited the ability to detect small or moderate effects. Future studies with larger cohorts are needed to confirm these preliminary findings

Finally, we included urologists who were not directly involved in the management of PCa (17%), which could have potentially diluted the results. However, most of them regularly attended multidisciplinary tumor board meetings, where PSMA-PET/CT imaging is routinely discussed. Their participation reflects the real-world heterogeneity of clinical exposure among urologists and was considered relevant to the assessment of broader interpretive skills within the urological community.

In the future, structured training programs for urologists should be developed to improve their PSMA-PET interpretation skills. In particular, the integration of artificial intelligence-based tools and the introduction of standardized reporting frameworks could further improve the diagnostic consistency [22]. In addition, it should be further investigated whether prolonged engagement with PSMA-PET cases within MDTs leads to measurable improvements in staging accuracy over time.

## 5. Conclusions

Our study highlights the complexity of PSMA-PET interpretation among urologists, particularly in T and M staging. Institutional experience, case volumes, and MDT attendance appear to influence diagnostic competence, emphasizing the need for structured training and interdisciplinary collaboration. As PSMA-PET becomes more widely used in the treatment of prostate cancer, improving urologists’ interpretation skills will be critical in optimizing patient outcomes.

## Figures and Tables

**Table 1 cancers-17-02122-t001:** Demographic and institutional characteristics of the 63 respondents, along with their proficiency (defined as ≥75% correct answers) in PSMA-PET-CT interpretation. Proficiency metrics include the correct identification of the local tumor stage (T stage), regional lymph node metastases (N stage), distant metastases (M stage), overall TNM staging (T, N, and M all correct), and partial staging accuracy (at least two out of three correct: T, N, or M within the same case).

Characteristic	Overall, N = 63 ^1^
**Age**	36 (30, 42)
**Hierarchy**	
Consultant	37 (59%)
Resident	26 (41%)
**Institution**	
Academic	39 (62%)
Non-academic	24 (38%)
**N° of PCa diagnoses per year**	
<300	18 (29%)
≥300	45 (71%)
**N° of RARPs per year**	
<150	35 (56%)
≥150	28 (44%)
**Personal experience in prostate biopsy**	43 (68%)
**Personal experience in fusion prostate biopsy**	31 (49%)
**Involvement in diagnosis and management of PCa**	52 (83%)
**PCa multidisciplinary team (MDT) presence**	40 (63%)
**Correct T stage**	11.0 (10.0, 15.0)
**Correct N stage**	14.00 (13.0, 16.00)
**Correct M stage**	12.00 (11.00, 14.00)
**Correct cases**	5.0 (4.0, 7.0)
**Almost * correct cases**	14.00 (13.0, 16.00)
**T stage proficiency ****	19 (30%)
**N stage proficiency ****	30 (48%)
**M stage proficiency ****	10 (16%)
**Correct case proficiency ****	1 (1.6%)
**Almost * correct case proficiency ****	28 (44%)

^1^ Median (IQR); n (%). * Almost correct cases = at least 2 correct identifications among T, N, and M stages; ** proficiency = at least 15/20 (75%) correct answers.

**Table 2 cancers-17-02122-t002:** Scores and proficiency (defined as reaching 75% of correct answers) in local staging (T stage), identification of regional lymph node metastases (N stage), identification of distant metastases (M stage), correct case staging (TNM staging), and almost correct case staging (at least 2 correct identifications among T, N, and M stages within the same case) in PSMA-PET-CT, stratified according to (A) type of institution (academic vs. non-academic), (B) number of PCa diagnoses per year per institution, (C) number of RARPs per year per institution, (D) presence of a prostate cancer multidisciplinary team in the institution, and (E) hierarchy (resident vs. consultant).

(A) Characteristic	Academic Institution, N = 39 (62%) ^1^	Non-Academic Institution, N = 24 (38%) ^1^	*p*-Value ^2^
**Correct T stage**	12 (10, 15)	11 (9, 16)	0.6
**Correct N stage**	14 (13, 16)	15 (13, 16)	0.3
**Correct M stage**	12 (12, 14)	12 (10, 14)	0.4
**Correct cases**	5 (4, 7)	5 (2, 10)	0.3
**Almost correct cases**	14 (13, 16)	14 (11, 18)	0.8
**T stage proficiency**	11 (28%)	8 (33%)	0.7
**N stage proficiency**	16 (41%)	14 (58%)	0.2
**M stage proficiency**	5 (13%)	5 (21%)	0.5
**Correct case proficiency**	0 (0%)	1 (4.2%)	0.4
**Almost correct case proficiency**	17 (44%)	11 (46%)	0.9
**(B) Characteristic**	**<300 PCA Diagnoses,** **N = 18 (29%) ^1^**	**≥300 PCA Diagnoses,** **N = 45 (71%) ^1^**	***p*-Value ^2^**
**Correct T stage**	11 (9, 13)	11 (10, 16)	**0.032**
**Correct N stage**	13 (12, 16)	15 (13, 16)	0.11
**Correct M stage**	12 (11, 12)	13 (12, 14)	**0.002**
**Correct cases**	4 (2, 6)	6 (4, 9)	**0.008**
**Almost correct cases**	13 (11, 15)	14 (13, 17)	0.11
**T stage proficiency**	2 (11%)	17 (38%)	**0.037**
**N stage proficiency**	6 (33%)	24 (53%)	0.2
**M stage proficiency**	0 (0%)	10 (22%)	**0.050**
**Correct case proficiency**	0 (0%)	1 (2.2%)	>0.9
**Almost correct case proficiency**	6 (33%)	22 (49%)	0.3
**(C) Characteristic**	**N° of RARPs per Year <150,** **N = 35 (56%) ^1^**	**N° of RARPs per Year ≥150,** **N = 28 (44%) ^1^**	***p*-Value ^2^**
**Correct T stage**	12 (9, 15)	11 (10, 16)	0.6
**Correct N stage**	13 (13, 16)	15 (13, 16)	0.13
**Correct M stage**	12 (11, 13)	13 (12, 14)	**0.030**
**Correct cases**	5 (3, 6)	6 (4, 10)	**0.042**
**Almost correct cases**	14 (13, 16)	14 (12, 18)	0.6
**T stage proficiency**	10 (29%)	9 (32%)	0.8
**N stage proficiency**	13 (37%)	17 (61%)	0.063
**M stage proficiency**	4 (11%)	6 (21%)	0.3
**Correct case proficiency**	0 (0%)	1 (3.6%)	0.4
**Almost correct case proficiency**	16 (46%)	12 (43%)	0.8
**(D) Characteristic**	**No MDT,** **N = 23 (37%) ^1^**	**MDT,** **N = 40 (63%) ^1^**	***p*-Value ^2^**
**Correct T stage**	12 (10, 15)	11 (10, 16)	>0.9
**Correct N stage**	14 (12, 16)	14 (13, 16)	0.7
**Correct M stage**	12 (11, 13)	13 (12, 14)	**0.026**
**Correct cases**	5 (3, 6)	5 (4, 7)	0.3
**Almost correct cases**	14 (12, 16)	14 (13, 16)	0.6
**T stage proficiency**	6 (26%)	13 (32%)	0.6
**N stage proficiency**	11 (48%)	19 (48%)	>0.9
**M stage proficiency**	2 (8.7%)	8 (20%)	0.3
**Correct case proficiency**	0 (0%)	1 (2.5%)	>0.9
**Almost correct case proficiency**	10 (43%)	18 (45%)	>0.9
**(E) Characteristic**	**Consultant,** **N = 37 (59%) ^1^**	**Resident,** **N = 26 (41%) ^1^**	***p*-Value ^2^**
**Correct T stage**	11 (10, 15)	11 (10, 15)	0.7
**Correct N stage**	15 (13, 16)	14 (12, 15)	0.090
**Correct M stage**	12 (11, 13)	13 (12, 14)	0.14
**Correct cases**	5 (3, 7)	5 (4, 7)	0.8
**Almost correct cases**	14 (13, 17)	14 (12, 16)	0.5
**T stage proficiency**	11 (30%)	8 (31%)	>0.9
**N stage proficiency**	21 (57%)	9 (35%)	0.083
**M stage proficiency**	6 (16%)	4 (15%)	>0.9
**Correct case proficiency**	1 (2.7%)	0 (0%)	>0.9
**Almost correct case proficiency**	17 (46%)	11 (42%)	0.8

^1^ Median (IQR); n (%). ^2^ Wilcoxon rank sum test; Pearson’s chi-squared test; Fisher’s exact test.

**Table 3 cancers-17-02122-t003:** Multivariable logistic regression analyses to assess predictors of achievement of proficiency (defined as 75% of correct answers) in (A) local disease (T stage), (B) regional lymph node metastases (N stage), (C) distant metastases (M stage), (D) almost correct case staging (at least 2 correct among T, N, and M staging).

(A) Local Disease	OR ^1^	95% CI ^1^	*p*-Value
**Hierarchy**			
*Consultant*	—	—	
*Resident*	0.91	0.27, 3.06	0.877
**N° of RARPs per year per institution**			
<150	—	—	
≥150	1.10	0.35, 3.40	0.873
**PCa MDT**			
*No*	—	—	
*Yes*	1.39	0.38, 5.32	0.624
**(B) Regional Lymph Node Metastases**	**OR ^1^ **	**95% CI ^1^ **	***p*-Value**
**Hierarchy**			
*Consultant*	—	—	
*Resident*	0.31	0.09, 1.02	0.062
**N° of RARPs per year per institution**			
<150	—	—	
≥150	2.98	0.99, 9.61	0.056
**PCa MDT**			
*No*	—	—	
*Yes*	1.212	0.338, 4.450	0.767
**(C) Distant Metastases**	**OR ^1^**	**95% CI ^1^**	***p*-Value**
**Hierarchy**			
*Consultant*	—	—	
*Resident*	0.61	0.13, 2.73	0.518
**N° of RARPs per year per institution**			
<150	—	—	
≥150	1.76	0.43, 7.89	0.440
**PCa MDT**			
*No*	—	—	
*Yes*	2.80	0.50, 22.10	0.266
**(D) A** **lmost correct case staging**	**OR ^1^**	**95% CI ^1^**	***p*-Value**
**Hierarchy**			
*Consultant*	—	—	
*Resident*	0.80	0.26, 2.47	0.702
**N° of RARPs per year per institution**			
<150	—	—	
≥150	0.86	0.30, 2.45	0.778
**PCa MDT**			
*No*	—	—	
*Yes*	1.22	0.37, 4.121	0.73

^1^ OR = odds ratio, CI = confidence interval.

## Data Availability

The data presented in this study are available in this article and Appendix A.

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
