# Peer review of "Interpretation of PSMA-PET Among Urologists: A Prospective Multicentric Evaluation"

_cancers, 2025, doi:10.3390/cancers17132122_

Round 1
Reviewer 1 Report
Comments and Suggestions for Authors
This multicenter prospective study evaluated the ability of 63 urologists from eight
Italian institutions to interpret PSMA-PET scans in high-risk prostate cancer (PCa)
cases. Better performance in identifying tumor (T) and metastasis (M) stages was
observed among urologists from high-volume centers and institutions with
multidisciplinary teams (MDTs). However, no significant predictors of proficiency
emerged in multivariable analysis.
The study addresses a clear gap and provides novel insights into proficiency levels in
PSMA-PET interpretation, highlighting institutional factors influencing staging
accuracy, which prior studies have not explored. Urologists' interpretation skills of
PSMA-PET have not been studied, despite its increasing clinical use.
The structure of the text is correct. The use of English is appropriate, although some
sentences require refinement (for example lines 113-115).
Introduction
In order to enhance the scientific rigor of the introduction, the authors should also
include the current guidelines of the international urological associations (European
or American) regarding the indications of PSMA-PET in the management of prostate
cancer.
Results
Line 118-120: It is apparent that there was variability in the correct staging
percentages across the 20 PSMA-PET cases, suggesting differences in case difficulty.
Therefore, the authors should provide a detailed breakdown of the correct staging
for each case in a separate table to enhance transparency and allow better
interpretation of the results.
Line 134-135: Please provide the corresponding p-value in the text.
Table 1
Please revise the table legend to improve readability.
Overall, this study makes a valuable contribution to the field by highlighting the
challenges urologists face in interpreting PSMA-PET scans for prostate cancer
staging. By tackling this issue, it paves the way for the development of structured
training programs and increased multidisciplinary engagement to improve staging
proficiency among urologists and ultimately enhance patient outcomes.
Author Response
Dear Reviewer 1,
Thank you for the positive feedback on the manuscript and for recognizing the clinical relevance of our study. Below, we address each of the Reviewer’s specific comments:
This multicenter prospective study evaluated the ability of 63 urologists from eight
Italian institutions to interpret PSMA-PET scans in high-risk prostate cancer (PCa)
cases. Better performance in identifying tumor (T) and metastasis (M) stages was
observed among urologists from high-volume centers and institutions with
multidisciplinary teams (MDTs). However, no significant predictors of proficiency
emerged in multivariable analysis.
The study addresses a clear gap and provides novel insights into proficiency levels in
PSMA-PET interpretation, highlighting institutional factors influencing staging
accuracy, which prior studies have not explored. Urologists' interpretation skills of
PSMA-PET have not been studied, despite its increasing clinical use.
The structure of the text is correct. The use of English is appropriate, although some
sentences require refinement (for example lines 113-115).
REPLY = We addressed these sentences by rephrasing in sentences in question.
Introduction
In order to enhance the scientific rigor of the introduction, the authors should also
include the current guidelines of the international urological associations (European
or American) regarding the indications of PSMA-PET in the management of prostate
cancer.
REPLY = We added this sentence to the introduction.
“According to current international guidelines, including those from the European Association of Urology (EAU) and the American Urological Association (AUA), PSMA PET/CT is recommended for staging in patients with unfavorable intermediate-, high-, and very high-risk prostate cancer, and for detection of biochemical recurrence after radical treatment when conventional imaging is inconclusive or negative”
Results
Line 118-120: It is apparent that there was variability in the correct staging
percentages across the 20 PSMA-PET cases, suggesting differences in case difficulty.
Therefore, the authors should provide a detailed breakdown of the correct staging
for each case in a separate table to enhance transparency and allow better
interpretation of the results.
REPLY = We added a detailed breakdown of the correct staging in Supplementary table 1
Line 134-135: Please provide the corresponding p-value in the text.
REPLY = This was corrected, sorry for missing to report it.
Table 1
Please revise the table legend to improve readability.
REPLY = the legend was rephrased to improve readability.
Overall, this study makes a valuable contribution to the field by highlighting the
challenges urologists face in interpreting PSMA-PET scans for prostate cancer
staging. By tackling this issue, it paves the way for the development of structured
training programs and increased multidisciplinary engagement to improve staging
proficiency among urologists and ultimately enhance patient outcomes.
Reviewer 2 Report
Comments and Suggestions for Authors
The article addresses an important and clinically valid topic: the ability of urologists to interpret PSMA PET scans. This fills a relevant gap in the literature and represents a clear strength of the manuscript. Remarks to the manuscript are as follows:
- Despite a solid hypothesis, most of the regression analyses did not yield statistically significant results. While the authors appropriately acknowledge this, it may raise questions about the statistical power or sample design.
- The 75% threshold for defining “proficiency” seems arbitrary and is not well justified in the text. The authors should consider providing a rationale for this cutoff or discussing alternative measures of competence.
- Heavy reliance on descriptive data - a substantial part of the results section is descriptive, with limited interpretative depth. The article would benefit from clearer clinical or training-related implications drawn from the data.
- Including a broader context and more cited sources in the discussion section, would enhance the generalizability and relevance of the findings.
Author Response
Dear Reviewer 2,
Thank you for the positive feedback on the manuscript and for recognizing the clinical relevance of our study. Below, we address each of the Reviewer’s specific comments:
- We acknowledge the Reviewer’s concern. As correctly noted, the absence of statistically significant associations in several regression models may reflect limited statistical power, particularly given the modest sample size. We have now added a brief discussion of this issue in the Limitations section, noting that the exploratory nature of the study and the available sample size may have limited the ability to detect small or moderate effects. Future studies with larger cohorts are needed to confirm these preliminary findings. (lines 258-262).
- Thank you for this important point. We agree that the choice of a 75% accuracy threshold may appear arbitrary. We selected this value as a commonly used benchmark in educational literature to indicate a minimum level of acceptable competence. We used the same cut-off in a previous publication with a similar endpoint (Mantica G, Suardi N, Smelzo S, Esperto F, Chierigo F, Tappero S, Borghesi M, La Rocca R, Oderda M, Ennas M, Stabile A, De Cobelli F, Napolitano L, Papalia R, Gontero P, Introini C, Briganti A, Scarpa RM, Mirone V, Montorsi F, Gaboardi F, Terrone C, Cardone G. Are Urologists Ready for Interpretation of Multiparametric MRI Findings? A Prospective Multicentric Evaluation. Diagnostics (Basel). 2022 Nov 1;12(11):2656. doi: 10.3390/diagnostics12112656. PMID: 36359499; PMCID: PMC9689928.). To address this, we discussed this choice in the Discussion, noting its limitations and suggesting that future research might explore alternative or more nuanced measures of proficiency (e.g., item-level difficulty analysis, expert consensus benchmarks, etc.) (lines 254-255).
- Thank you for this constructive comment. In the revised manuscript, we have streamlined the Results section by reducing the descriptive content, which is already clearly presented in the tables. We have also revised the Discussion to better interpret the results, particularly with regard to their clinical and educational implications.
- Thank you for this suggestion. We have expanded the Discussion section to place our findings within a broader context. We now reference recent literature on the challenges of imaging interpretation in non-radiologist specialties, and the emerging need for multidisciplinary training.
Reviewer 3 Report
Comments and Suggestions for Authors
Compliments to the authors for a multicenter assessment of the urologists capability at correctly interpreting PSMA PET scans. They have also assessed the factors that can play a role in improving their proficiency at this reading.
Few small comments:
- Line 86-87: you have written, No clinical data was given to the readers in order not to influence the outcomes of the PSMA-PET lecture; do you mean lecture or interpretation?
- Have you defined the experience of the consultants (how many years in practice?)?
- Why include in this study urologists not directly involved in the diagnosis and management of PCa? This would dilute the results of the study. It would be obvious that people whose area of interest is not cancer prostate would not have any proficiency in interpreting PSMA PET.
- Line 114: you have mentioned, 68% had experience in prostate biopsy, and 49% In fusion prostate biopsy; I suggest, 68% had experience in prostate biopsy, while most of them (49%) had an experience with MRI fusion biopsy.
- There is a lot of repetition of data between the ‘results’ and the tables. The results can be shortened. Whatever is not vital like demographics can be just mentioned in tables.
- In table 4 you have mentioned about the four ‘Characteristic’, instead better to specify what characteristic is being looked at. It would improve the understanding of the table.
Author Response
Dear Reviewer 3,
Thank you for the time and effort spent reviewing our manuscript.
Thank you for your positive feedback.
- We meant interpretation. We have edited the text accordingly.
- Although we did not explicitly report the number of years in independent clinical practice, all consultants included in the study had at least 3 years of post-recidency. We have now included this clarification in the revised manuscript (line 83).
- We understand the concerns regarding the involvement of urologists not directly involved in the management of PCa. However, most of the participants (83%) were directly involved in the diagnosis and treatment of PCa. The remaining participants were indirectly involved in PCa treatment by attending multidisciplinary tumour board meetings where PSMA PET/CT imaging is routinely discussed. Moreover, their participation reflects the real-world heterogeneity of clinical exposure among urologists and was considered relevant to the assessment of broader interpretive skills within the urological community. To address your point, we have revised the manuscript to clarify these aspects and added a brief note in the limitations section discussing this (lines 252-258).
- Thank you for your suggestion. We have revised the sentence accordingly for improved clarity. The new sentence now reads: “In total, 68% of participants had experience in prostate biopsy, while most of them (49%) had experience with MRI fusion biopsy.”
- We appreciate this observation. In the revised manuscript, we have shortened the text in the Results section by reducing redundancy with the tables. Demographic details and institutional characteristics are now primarily presented in the tables.
- Thank you for pointing this out. We have revised the column headers in Table 4 to clearly specify the outcomes being evaluated.
Round 2
Reviewer 3 Report
Comments and Suggestions for Authors
Compliments to the authors for a very novel study looking at the proficiency of the interpretation of PSMA PET among urologists.
The corrections and changes are acceptable and good.